# Leveraging Prediction Inconsistency for Online Error Detection in Procedural Videos

## Abstract

An efficient and accurate system for detecting errors in procedural tasks is crucial for supporting human needs in daily life, as it can provide instant notifications and guide people to correct mistakes. In this paper, we address the challenge of real-time and online error detection in procedural task videos by leveraging inconsistencies in action detector predictions. We propose a DUal-Branch Action Detector (DUBAD) framework, which integrates both *robust* and *sensitive* actions detectors. The *robust* action detectors generate accurate and stable action predictions, while the *sensitive* detectors produce inconsistent predictions when errors occur. To achieve this, we design a temporal-aware dynamic weight module that enhances sensitivity to errors using affine transformations with input-dependent, constrained weights and biases. Furthermore, we train the action detectors with varying amounts of temporal information to amplify inconsistencies in prediction when action sequences deviate from the correct order. For videos containing multiple or diverse errors, we apply a majority voting scheme based on mismatches between robust and sensitive predictions. Extensive experiments on EgoPER, Assembly-101-O, and EPIC-Tent-O demonstrate that our method outperforms state-of-the-art approaches in online error detection, while maintaining real-time efficiency with a lightweight architecture. [1]

## 1 Introduction

People have benefited greatly from advances in procedural video understanding across a wide range of problem domains and applications, including action detection (Eun et al. (2020); Wang et al. (2023b); Zhao & Krähenbühl (2022); An et al. (2023); Wang et al. (2023a); Pang et al. (2025)), segmentation (Lu & Elhamifar (2024); Shen & Elhamifar (2024); Lu & Elhamifar (2025); Farha & Gall (2019); Li et al. (2020); Yi et al. (2021); Souri et al. (2021); Li et al. (2022); Liu et al. (2023)), video grounding (Lu et al. (2025); Li et al. (2024); Mu et al. (2024); Dvornik et al. (2021; 2023); Ashutosh et al. (2023)), and error understanding (Wang et al. (2023c); Ghoddoosian et al. (2023); Ding et al. (2023); Flaborea et al. (2024); Lee et al. (2024); Luigi Seminara (2024); Huang et al. (2025)). Imagine preparing pasta with the assistance of an AI system that guides you through each step. If you make a mistake, the assistant provides instant feedback, enabling you to correct the error and continue seamlessly. For such assistance to be useful, error detection must be both real-time and online, operating only on past frames as people perform tasks sequentially. Meanwhile, these AI assistants are often deployed on wearable devices such as Microsoft HoloLens or Apple Vision Pro. These devices provide a natural first-person view and enable hands-free operation, but they also impose strict efficiency and latency constraints. Thus, *development of a real-time and online error detection system for egocentric procedural videos* is crucial for enhancing both the user experience and task proficiency.

Recent studies have categorized errors into two types: execution errors and procedural errors( Flaborea et al. (2024); Lee et al. (2024); Luigi Seminara (2024); Huang et al. (2025)). Execution errors occur when actions deviate from the correct ones, interrupt the procedure, or introduce irrelevant steps. Examples include accidentally dropping a knife while preparing food, adding sugar instead of honey, or inserting oats into a tortilla during quesadilla preparation. In contrast, procedural errors

---

[1]Code will be released upon acceptance.

involve repeating, omitting, or misordering steps. For instance, tightening all table legs before inserting a cross support violates the correct assembly sequence. Execution errors are well captured in datasets like EgoPER (Lee et al. (2024)) and CaptainCook4D (Peddi et al. (2024)), while procedural errors are the focus of PREGO (Flaborea et al. (2024)), derived from Assembly-101 (Sener et al. (2022)) and EPIC-Tent (Jang et al. (2019)).

Existing methods tackle these errors either offline or online. Offline methods (Lee et al. (2024); Huang et al. (2025)) require full videos and jointly perform temporal action segmentation and error detection. Online methods, in contrast, process current and past frames causally. Both largely adopt the One-Class Classification (OCC) paradigm, training only on normal (erro-free) videos. Although recent online methods show strong benchmark performance, they suffer from key shortcomings: (1) they only detect the first error in a video, limiting real-world applicability where multiple mistakes are common such as in EgoPER (Lee et al. (2024)) and CaptainCook4D (Peddi et al. (2024)); and (2) they mainly address procedural errors while overlooking nuanced execution errors as observed in our experiments. For example, PREGO (Flaborea et al. (2024)) relies on discrepancies between Large Language Model (LLM)-based anticipation and recognition labels, which hinders real-time performance and assumes correctness of prior predictions. DTGL (Seminara et al. (2024)) flags deviations from ground-truth task graphs but fails to capture execution errors, where actions appear in sequence but are incorrectly performed. This leaves a critical gap for real-world assistive applications.

To address these limitations, we propose DUal-Branch Action Detector (DUBAD), a light-weight framework for online error detection that captures both procedural and execution errors. Our framework is an ensemble of action detectors based on Recurrent Neural Networks (RNNs), trained only on normal (error-free) videos. At inference time, we exploit prediction inconsistencies among these detectors for error detection. *The detectors differ in their temporal receptive fields and sensitivity to input data.* This design choice ensures consistent predictions for normal/correct actions, while producing conflicting outputs in the presence of unseen or erroneous cases. Inspired by miniROAD (An et al. (2023)), which studies the effect of varying amounts of temporal information for action detection, we design two model branches that differ in their temporal receptive fields, introducing diversity in how much past context influences predictions. Within each branch, we implement a *robust* and *sensitive* action detector. The sensitivity of the action detector to in-distribution data (i.e., normal videos) influences its prediction behavior under distributional shifts. Therefore, to make the action detector more robust to out-of-distribution data (i.e., erroneous videos), we propose a Step Attention Module (SAM) to augment input frames using action prototypes as additional input to train a *robust* action detector. In addition, to increase data sensitivity, we introduce a Temporal Aware Dynamic (TAD) module in our sensitive action detector. TAD generates temporal-aware and input-dependent weights and biases for the affine transformation applied to each frame feature after SAM. The resulting features for the *sensitive* action detector tend to produce unstable or inconsistent action predictions when errors occur. Our contributions are summarized as follows:

– We introduce DUBAD, a lightweight framework that explicitly leverages prediction inconsistency for real-time online error detection.

– We design two complementary detectors: (i) a *robust* branch with a Step Attention Module (SAM) for stable predictions, and (ii) a *sensitive* branch with a Temporal-Aware Dynamic (TAD) module that amplifies inconsistencies under errors. Training with different temporal receptive fields further diversifies prediction behaviors. The design enables our proposed framework to detect both procedural and execution errors.

– We achieve state-of-the-art performance on Assembly-101-O, EPIC-Tent-O, and EgoPER, demonstrating that DUBAD effectively detects both execution and procedural errors in real time with parameter efficiency.

## 2 RELATED WORK

**Online Action Detection.** Prior works (Eun et al. (2020); Wang et al. (2023b); Zhao & Krähenbühl (2022); An et al. (2023); Wang et al. (2023a); Pang et al. (2025)) have widely studied Online Action Detection (OAD), one of the popular directions in procedural video understanding. Given a video containing multiple actions, an OAD model identifies the actions taking place using only past and

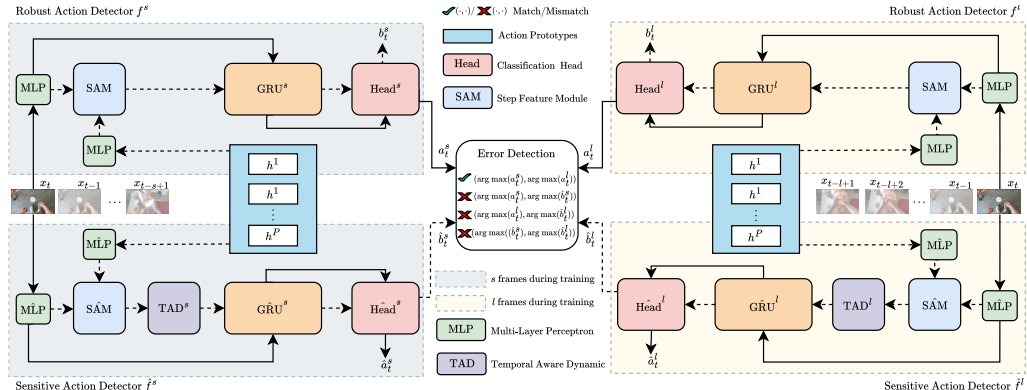

Figure 1: The pipeline of our framework DUal-Branch Action Detector (DUBAD). The *robust* action detectors $f^s$ and $f^l$ trained with $s$ and $l$ frames produce action predictions $a_t^s$ and $a_t^l$ (solid line). Similarly, the *sensitive* action detectors $\hat{f}^s$ and $\hat{f}^l$ produce $\hat{b}_t^s$ and $\hat{b}_t^l$ (dashed line). During inference for error detection, we construct four pairs among $a_t^s, a_t^l, \hat{b}_t^s$, and $\hat{b}_t^l$; if three pairs mismatch, the corresponding frame is flagged as an error. Solid and dashed lines represent two different inputs, respectively.

current frames. Specifically, miniROAD (An et al. (2023)) is built based on RNNs and adopts selected weights to only train the last frame of a given sequence of frames to alleviate the discrepancy between training and inference. The design is simple yet effective for addressing OAD. A recent work, CMeRT (Pang et al. (2025)), further alleviates training-inference discrepancy in previous OAD methods due to imbalanced context exposure in long- and short-term memory. It adopts a context-enhanced encoder using near-past context to learn consistent short-term representations, and a memory-refined decoder uses near-future context with learned short-term representations to detect actions. We construct our proposed framework based on miniROAD and its training loss.

**Error Detection in Procedural Videos.** The research community has recently shown growing interest in error detection, which aims to detect the occurrence of or localize an action that changes the action sequence of a procedure or should not occur in a procedure. Specifically, among the offline methods, EgoPED (Lee et al. (2024)) determines the execution errors by thresholding the similarities between the predicted action features and input frames features. AMNAR (Huang et al. (2024)) generates action features conditioned on executed actions and follows the same strategy as in EgoPED for error detection. On the other hand, among the online methods, PREGO (Flaborea et al. (2024)) detects procedural errors by the difference in prediction generated by an action detector and LLM, with its strong reasoning capability to anticipate the next action given past actions. DTGL (Seminara et al. (2024)) learns the task graph from the training videos and detects an action as a procedural error if its preconditions in the learned task graph are not in the observed actions. However, existing online methods fail to handle execution errors or videos with multiple errors.

## 3 PROPOSED METHOD

### 3.1 PROBLEM SETTING AND FRAMEWORK OVERVIEW

Given a frame feature $x_t$ at time $t$ in a video with length $T$ and past frame features $(x_0, x_1, \ldots, x_{t-1})$, we aim to predict $y_t \in \{0, 1\}$, where $x_t \in \mathbb{R}^D$ and $y_t = 1$ indicates an error occurs in frame $x_t$, otherwise $y_t = 0$. This procedure starts from the first frame and continues until the end of the video. Note that all training videos are error-free and only the ground-truth frames-wise actions $(\bar{y}_0, \bar{y}_1, \ldots, \bar{y}_T)$ are available, where $\bar{y}_t \in \{1, 2, \ldots, P + 1\}$, $P + 1$ is the number of actions, and $\bar{y}_t = P + 1$ denotes background.

Our proposed DUal-Branch Action Detector (DUBAD) is illustrated in Fig. 1. It consists of two branches, each of which contains two RNN-based action detectors. One of the branches consists of the *robust* and *sensitive* action detectors $f^s$ and $\hat{f}^s$ trained with $s$ frames. The other branch consists of the *robust* and *sensitive* action detector $f^l$ and $\hat{f}^l$ trained with $l$ frames. The *robust* action detector contains the proposed Step Attention Module (SAM) while the *sensitive* action detector is made of

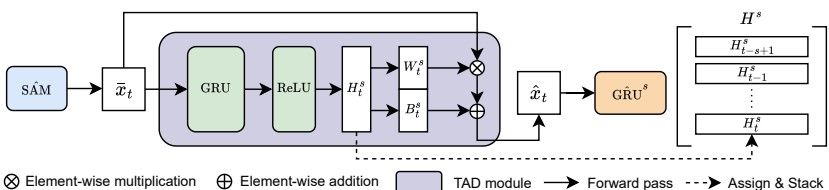

Figure 2: Pipeline of the TAD module with $s$ frames. The TAD module with $l$ frames uses the same pipeline.

SAM followed by our Temporal Aware Dynamic (TAD) module. During inference, four predictions from the two branches are combined into four comparison pairs, and an error is detected if at least three of the pairs mismatch (middle of Fig. 1). For simplicity, we describe the forward training process of the *robust* and *sensitive* action detectors $f^s$ and $\hat{f}^s$.

## 3.2 ROBUST ACTION DETECTOR WITH STEP ATTENTION MODULE (SAM)

We aim to develop action detectors robust to erroneous frames, enabling them to consistently produce accurate and stable predictions. To begin (see the solid arrow in the upper-left region of Fig. 1), the *robust* action detector of the short range branch $f^s$ processes a sequence of $s$ frame features $X_t^s = (x_{t-s+1}, ..., x_{t-2}, x_{t-1}, x_t)$ ending at time $t$ through a multi-layer perceptron (MLP), Gated-Recurrent Unit (GRU), and a linear layer as the classification head to predict the action logits $a_t^s \in \mathbb{R}^{P+1}$ at time $t$, where $a^s \in \mathbb{R}^{T \times P+1}$ denoting action logits after processing the entire video.

To enhance the robustness of the action detector, i.e., $\text{GRU}^s$ and $\text{Head}^s$, especially when execution errors occur, we propose SAM to augment the features using action prototypes after MLP as an additional training input for the action detector (see the dashed arrow in the upper-left region of Fig. 1). The action prototype encodes the semantics of the corresponding action, guiding the detectors to learn generalizable patterns for action detector and produce stable predictions under errors. First, we generate the action prototype $h^i \in \mathbb{R}^D$ for each action $i \in \{1, 2, \ldots, P\}$ by averaging over frame features corresponding to the action $i$ in all training videos. We exclude background frames because background features are too diverse to be averaged as a prototype for feature matching. Second, we use a multi-head attention module (Vaswani et al. (2017)) to generate augmented frame features $\bar{X}_t^s$ corresponding to all frames of $X_t^s$, where the *query* is the frame feature $x \in X_t^s$, the *key* and *value* are the action prototypes $h \in \mathbb{R}^{P \times D}$, where each row $i$ corresponds to $h^i$. See our supplementary material for implementation details. Finally, we feed $\bar{X}_t^s$ to $\text{GRU}^s$ and $\text{Head}^s$ to predict the action logits $b_t^s \in \mathbb{R}^{P+1}$ at time $t$ with $b^s \in \mathbb{R}^{T \times P+1}$ representing action logits after processing the entire video. We follow this forward training process for $f^l$ with $l$ frames to obtain $\bar{X}_t^l, a_t^l$, and $b_t^l$. Notice that the difference in the input temporal range introduces inconsistencies in the predictions of the two branches when actions do not follow their expected sequences during inference.

## 3.3 SENSITIVE ACTION DETECTOR WITH TEMPORAL AWARE DYNAMIC (TAD) MODULE

Training with the augmented frame features $\bar{X}_t^s$ and $\bar{X}_t^l$ from SAM increases the robustness of action detectors. Next, we build the *sensitive* action detectors $\hat{f}^s$ and $\hat{f}^l$ that predict inconsistent action classes, especially when execution errors occur. Inconsistencies in predictions of different action detectors enable error detection. To establish sensitivity (see the dashed arrow in the lower-left region of Fig. 1), we build $\hat{f}^s$ based on $f^s$ by additionally proposing the TAD module to convert $\bar{X}_t^s$ and $\bar{X}_t^l$ into temporal-aware representations that are more sensitive to erroneous frames. Fig. 2 shows the $\text{TAD}^s$ module, consisting of a GRU, affine transformation, and the resulting temporal-aware weight and bias vector $H_t^s \in \mathbb{R}^{2D}$. We take $\bar{X}_t^s$ as input for the GRU to generate $H_t^s$ for time $t$. We divide $H_t^s$ into two parts: $W_t^s$ as the weight and $B_t^s$ as the bias, where $W_t^s, B_t^s \in \mathbb{R}^D$. Following that, an affine transformation is applied to the augmented frame feature $\bar{x}_t \in \bar{X}_t^s$ to obtain representation $\hat{x}_t^s = W_t^s \bar{x}_t + B_t^s$. We then feed $\hat{X}_t^s = (\hat{x}_{t-s+1}^s, ..., \hat{x}_{t-1}^s, \hat{x}_t^s)$ through the $\hat{\text{GRU}}^s$ and $\hat{\text{Head}}^s$ to output $\hat{b}_t^s$ only for the current frame $t$. Meanwhile, the *sensitive* action detector also takes $x_t$ as an input to predict $\hat{a}_t^s$ to stabilize the training. Similar to Section 3.2, in order to introduce

temporal variations in training, we follow the same forward process with $\hat{f}^l$ to obtain $H_t^l$, $\hat{X}_t^l$, $\hat{a}_t^l$, and $\hat{b}_t^l$ with $l$ frames, where $\hat{x}_t^l = W_t^l \bar{x}_t + B_t^l$.

## 3.4 TRAINING LOSSES

DUBAD is trained on batches of frame sequences, each independently and randomly sampled from the training videos. Concretely, each batch consists of training pairs $(v, t)$ from a random time $t$ and video $v$. Next, we describe the training losses applied to the *robust* and *sensitive* action detectors.

**Classification Loss.** We adopt the same cross-entropy loss as in miniROAD (An et al. (2023)). The classification loss is computed only for the frame at time $t$ while observing its $k \in \{s, l\}$ prior frames. Additionally, we utilize a smoothing loss that minimizes the difference between two consecutive predictions in time. In particular, for time $t$ of video $v$, let $\bar{y}_{v,t}$ be the ground-truth action label, and $p_{v,t}^k \in \mathbb{R}^{P+1}$ be the softmax values computed from the action logits of any of the four classification heads of branch $k$ such as $a^k$ or $b^k$. The two losses for each sample in the batch are formulated below. For simplicity, we omit the video index $v$ hereafter:

$$\mathcal{L}_{\text{CE}}^k(t) = -\sum_{p \in \{a,b,\hat{a},\hat{b}\}} \log(p_{t,\bar{y}_t}^k), \tag{1}$$

$$\mathcal{L}_{\text{Smo}}^k(t) = \sum_{p \in \{a,b,\hat{a},\hat{b}\}} \frac{1}{P+1} \sum_{i=1}^{P+1} \left(\log p_{t,i}^k - \log p_{t-1,i}^k\right)^2. \tag{2}$$

For each sample in the batch, the total classification loss $L_{\text{cls}}^k(t) = L_{\text{CE}}^k(t) + L_{\text{Smo}}^k(t)$ is defined as the sum of the two losses for both the short and long range branches.

**Temporal Aware Dynamic Loss.** This is an auxiliary loss which is specifically designed to support the training of the *sensitive* action detectors $\hat{f}^s$ and $\hat{f}^l$. As shown in Fig. 2, in order to compute the TAD loss, we formulate $H^s$ and $H^l$ by stacking $(H_{t-s+1}^s, H_{t-1}^s, \ldots, H_t^s)$ and $(H_{t-l+1}^l, H_{t-1}^l, \ldots, H_t^l)$ in memory for the short and long range branches respectively. This is different from the cross-entropy loss where the loss is only computed for the frame $t$. Consequently, we define the TAD loss for each sample in the batch as follows:

$$\mathcal{L}_{\text{TAD}}^k(t) = \max\left(0, \frac{1}{2D} \sum_{i=0}^{k-1} \left((H_{t-i}^k)^\mathsf{T} (H_{t-i}^k)\right) - \beta\right), \tag{3}$$

where $D$ is the feature dimension, $k \in \{s, l\}$, and $\beta$ is the margin that controls the sparsity of $H^k$. Eq. 3 encourages $H^s$ and $H^l$ to select key semantic features of the frames. Furthermore, we use the same margin for the two temporal branches, i.e., $H^l$ and $H^s$. As $H^l$ incorporates more frames, it tends to choose fewer semantic features than $H^s$, leading to a different extent of error sensitivity for $\hat{f}^l$. Overall, $\mathcal{L}_{\text{TAD}}^k$ utilizes the temporal-aware and input-dependent design of the TAD to make the weights $H^k$ more sensitive to unseen error instances during inference.

**Final Batch Training.** To conclude, the losses of the two branches are added to derive the final loss $\mathcal{L}_{\text{final}}(t) = \sum_{k \in \{s,l\}} \mathcal{L}_{\text{TAD}}^k(t) + \mathcal{L}_{\text{cls}}^k(t)$ which is further averaged over all sequences in the batch to train the *robust* and *sensitive* detectors.

## 3.5 ONLINE ERROR DETECTION DURING INFERENCE

At inference time, we process the video frame by frame from $t = 1$ to $t = T$ to obtain predictions from the action detectors. We then 1) apply a causal mode filter, which only uses past frames to replace the action with the most frequent action within a window size $m$ to smooth the predictions, and 2) perform error detection via *action prediction inconsistency*. First, we capture the inconsistencies in the following pairs: $(a_t^s, \hat{b}_t^s)$ and $(a_t^l, \hat{b}_t^l)$, where $a_t^s$ and $a_t^l$ provide correct or specific action predictions on errors, resulting in stable outputs. Meanwhile, $\hat{b}_t^s$ and $\hat{b}_t^l$ provide inconsistent action predictions on errors due to sensitive frame features $\hat{X}_t^s$ and $\hat{X}_t^l$. Inconsistencies in the two pairs

Table 1: Online error detection performance on EgoPER, Assembly-101-O, and EPIC-Tent-O.

| Method | EgoPER | | | Assembly-101-O | | | EPIC-Tent-O | | |
|---|---|---|---|---|---|---|---|---|---|
| | F1@10 | F1@25 | F1@50 | Avg-F1 | E-F1 | C-F1 | Avg-F1 | E-F1 | C-F1 |
| MSGI (Sohn et al. (2020)) | - | - | - | 33.1 | 43.5 | 22.7 | 44.5 | 22.0 | 66.9 |
| MSG$^2$ (Jang et al. (2023)) | - | - | - | 46.2 | 33.2 | 59.1 | 45.2 | 22.9 | 67.5 |
| PREGO (Flaborea et al. (2024)) | 41.2 | 28.1 | 12.2 | 32.5 | 41.8 | 23.1 | 29.4 | 17.2 | 41.6 |
| DTGL (Seminara et al. (2024)) | 39.6 | 33.8 | 20.9 | 53.5 | 28.1 | **78.9** | 46.5 | 23.7 | 69.3 |
| DUBAD (Ours) | **52.1** | **42.8** | **23.9** | **67.1** | **74.1** | 60.2 | **58.8** | **27.6** | **90.0** |

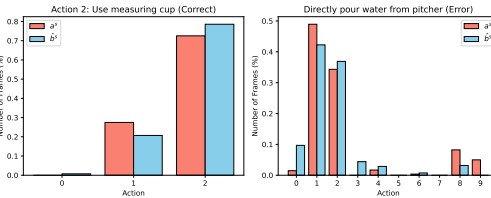
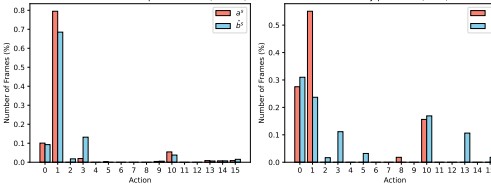

(a) Correct (left) and erroneous(right) unseen action 2.     (b) Correct (left) and erroneous(right) unseen action 1.

Figure 3: Histograms of predicted actions $a^s$ (red) and $\hat{b}^s$ (blue) for correct actions and the corresponding errors in *tea* (a) and *oatmeal* (b) of EgoPER. Y- and x-axis denote the action ratios and the indices of actions.

are valuable for capturing execution errors, since subtle variations often differentiate correct actions from errors. On the other hand, we form another two pairs: $(a_t^s, a_t^l)$ and $(\hat{b}_t^s, \hat{b}_t^l)$, generated by the action detectors trained with short and long temporal range ($s$ and $l$). Inconsistencies observed in the two pairs imply that the predicted action does not follow any correct action sequence and thus, facilitate the detection of procedural errors. Finally, in a real-world scenario, multiple types of errors can occur in one video and interactively influence each other. To finally determine whether a frame contains an error or not, we use majority voting on the mismatches for those four pairs. We flag a frame as an error ($\hat{y}_t = 1$) if three mismatches are detected where $\hat{y}_t \in \{0, 1\}$ is the final prediction.

## 4 EXPERIMENTS

### 4.1 EXPERIMENTAL SETUP

**Dataset.** We evaluate our proposed method on EgoPER (Lee et al. (2024)), Assembly-101-O (Flaborea et al. (2024)), and EPIC-Tent-O (Flaborea et al. (2024)). EgoPER consists of 5 tasks (*tea*, *quesadilla*, *oatmeal*, *oatmeal*, and *coffee*) with 386 egocentric videos and contains both execution and procedural errors. We use the given training/test split for evaluation. For Assembly-101-O and EPIC-Tent-O, which consist of egocentric videos with procedural errors in assembly domain, we follow the same training/test split as in PREGO (Flaborea et al. (2024)) and DTGL (Seminara et al. (2024)). Note that the procedure in each test video stops once an error occurs, meaning that only the last action is flagged as an error.

**Evaluation Metrics.** For EgoPER, we report the metric used in (Farha & Gall (2019)), the segment-wise F1 score under three overlap thresholds (10%, 25%, and 50%) based on normal and erroneous segments, denoted as F1@10, F1@25, and F1@50. They consider both localization and classification performance of error detection, especially for execution errors. For Assembly-101-O and EPIC-Tent-O, we report the same action-wise F1 scores as in DTGL for two complementary classification cases: erroneous → correct (E-F1) and correct → erroneous (C-F1). Their average yields Avg-F1. All F1 scores are computed based on predicted actions.

**Baselines.** We mainly compare our proposed method with two state-of-the-art online error detection methods, PREGO (Flaborea et al. (2024)) and DTGL (Seminara et al. (2024)) with direct optimization which performs the best in general. For EgoPER, we train PREGO with Qwen 2.5 (Yang et al. (2024)) as its LLM and DTGL from scratch with their public codes and evaluate their performance.

Table 2: Inference speed analysis using frame duration (FD) in seconds (s) and frame per second (FPS).

| Method | Feature Extractor | | Error Detection | | | Combined | |
|---|---|---|---|---|---|---|---|
| | FD (s) | FPS | FD (s) | FPS | Parameters | FD (s) | FPS |
| PREGO (Flaborea et al. (2024)) | 0.0261 | 38.2 | 0.7230 | 1.38 | $\sim$ 14B | 0.749 | 1.33 |
| DTGL (Seminara et al. (2024)) | 0.0261 | 38.2 | 0.0008 | 1180 | 5.72M | 0.027 | 37.0 |
| DUBAD (Ours) | 0.0261 | 38.2 | 0.0148 | 67.5 | 18.8M | 0.041 | 24.4 |

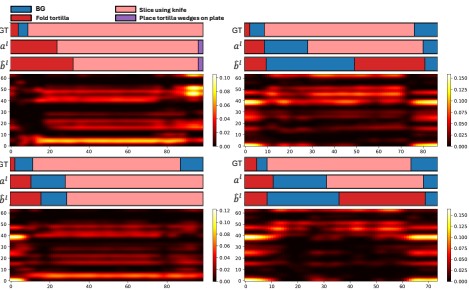
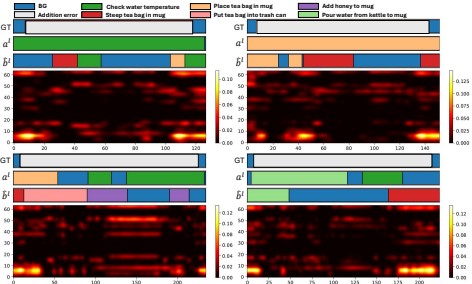

(a) Left column represents correct actions. Right column represents erroneous actions for *quesadilla*.

(b) All four figures denote the same error but from different videos for *tea*.

Figure 4: Each row in a sub-figure, from top to bottom shows frame-wise ground-truth (GT) actions, $a^l$, $\hat{b}^l$, and the heatmap of $\hat{X}^l$ for errors on *quesadilla* (a) and *tea* (b) of EgoPER. For $\hat{X}^l$, x-axis represents the number of channels (we only visualize 64 of them) and y-axis denotes the number of frames.

For Assembly-101-O and EPIC-Tent-O, we include the performance of PREGO, DTGL, and other two baselines, MSGI (Sohn et al. (2020)) and MSG$^2$ ((Jang et al. (2023))) reported in DTGL.

**Implementation Details.** For EgoPER, we use TimeSFormer (Bertasius et al. (2021)), pre-trained on Ego4D (Grauman et al. (2021)), to extract frame-level features using 4 past frames as input at 10 frames per second (FPS). For Assembly-101-O and EPIC-Tent-O, we use the frame features generated by PREGO at 30 and 60 FPS, respectively. The MLP consists of a fully-connected layer, a Layer Normalization layer, and ReLU. Every GRU has one layer and every classification head has one fully-connected layer. The modules in Fig. 1 with the same name share the same weight. We train the model using AdamW optimizer with learning rate 0.0001 and weight decay 0.05 for 20 epochs. To perform error detection on Assembly-101-O and EPIC-Tent-O, we output the error probability $\hat{y}$ and extract a sequence of actions from $a^s$ because it has relatively stable and accurate results in general. We flag the action as an error if any frame within the segment is detected as an error in $\hat{y}$. In addition, to prevent overfitting, we uniformly sample one frame for every four frames in Assembly-101-O and every six frames in EPIC-Tent-O during training and inference, respectively. See our supplementary material for hyperparameter ($s$, $l$, $\beta$, and $m$) setting. We train and evaluate all models using Intel Xeon(R) Silver 4210 CPU@2.20GHz and NVIDIA RTX A6000.

### 4.2 EXPERIMENTAL RESULTS

**Online Error Detection.** Table 1 compares error detection performance between our method (DUBAD) and baselines on EgoPER, Assembly-101-O, and EPIC-Tent-O. DUBAD outperforms all baselines, specifically achieving 52.1%, 42.8%, and 23.9% on F1@10, F1@25, and F1@50 compared to 39.6%, 33.8%, 20.9% by DTGL on EgoPER. The results show that in complex scenarios with procedural and execution errors, DUBAD detects errors more effectively than the baselines. On the other hand, DUBAD achieves 13.6% and 12.3% higher Avg-F1 scores than DTGL on Assembly-101-O and EPIC-Tent-O, respectively. The higher Avg-F1 score indicates that DUBAD produces consistent predictions for normal actions and inconsistent predictions when procedural errors occur.

**Inconsistency Analysis in Prediction.** Fig. 3 shows the histogram of frame-wise *robust* and *sensitive* predictions ($a^s$ and $\hat{b}^s$) with regards to two correct actions and their corresponding unseen errors in the EgoPER dataset. In the left figures of Fig. 3 (a) and (b), the *robust* and *sensitive* action

Table 3: Ablation study of DUBAD. RGB refers to the pipeline shown by the solid line in Fig. 1.

| $f^s$ and $f^l$ | | $\hat{f}^s$ and $\hat{f}^l$ | | | EgoPER | | | EPIC-Tent-O | | |
|---|---|---|---|---|---|---|---|---|---|---|
| RGB | SAM | RGB | SAM | TAD | F1@10 | F1@25 | F1@50 | Avg-F1 | E-F1 | C-F1 |
| ✓ | | ✓ | ✓ | | 47.8 | 37.6 | 22.6 | 53.7 | 17.1 | **90.2** |
| ✓ | ✓ | ✓ | ✓ | | 47.7 | 38.1 | **24.2** | 53.5 | 19.5 | 87.5 |
| ✓ | ✓ | ✓ | ✓ | ✓ | **52.1** | **42.8** | 23.9 | **58.8** | **27.6** | 90.0 |

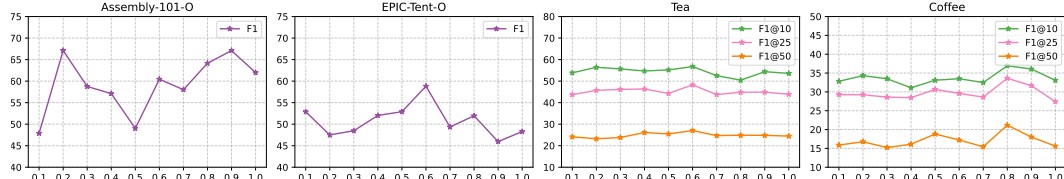

Figure 5: Performance of different margins $\beta$. Y-axis and x-axis denote F1 scores and margins, respectively.

detectors ($f^s$ and $\hat{f}^s$) demonstrate similar diversity and prediction pattern for correct action 2 during making *tea* and action 1 during making *oatmeal* of EgoPER. Meanwhile, the right figures of Fig. 3 (a) and (b) show that $\hat{f}^s$ produces more inconsistent predictions than $f^s$ for erroneous actions. Specifically, as seen in the right side of Fig. 3 (b), predictions for the *sensitive* detector $\hat{f}^s$ are split among 8 different classes while the *robust* detector $f^s$ predicts only 4 actions for the erroneous action of "directly pouring oats".

**Complexity Analysis.** Table 2 shows the comparison between different methods in terms of inference speed and model size. Note that the combined setting better reflects real-world scenarios as it considers both frame processing and error detection. DUBAD achieves real-time processing (24.4 FPS) in the combined setting, and outperforms PREGO, which attains only 1.33 FPS using LLMs. On the other hand, DTGL pre-computes the task graph, enabling no-delay error detection through pre-condition matching and only requiring a lightweight action detector (5.72M). Although DTGL obtains a higher FPS with fewer parameters than DUBAD, the latter surpasses the former in F1 scores across three different datasets while maintaining real-time processing speed and parameter efficiency (18.8M).

**TAD Module Analysis.** We investigate the features generated by the TAD module. To enhance contrast and highlight patterns, we visualize $\hat{X}_t^l$ due to its high feature sparsity. Here, brighter colors indicate higher values in the feature maps. The figures in the left column of Fig. 4 (a) correspond to the correct action "slice tortilla using knife" for two different videos. The action predictions $a^l$ and $\hat{b}^l$ as well as corresponding $\hat{X}^l$ are similar. In contrast, the figures in the right column of Fig. 4 (a) refer to the erroneous action "tear tortilla into half *by hand*" instead of "*using knife*". $\hat{X}^l$ for erroneous instances inconsistently shows different semantic features from the one for the correct action. This leads to unexpected predictions, i.e, "fold tortilla" (marked in red), for $\hat{b}^l$ compared to those of $a^l$. Notice $a^l$ consistently predicts the closest action, i.e., "slice tortilla". On the other hand, the four figures in Fig. 4 (b) represent the same unseen action "put mug into microwave for 5 seconds" from different videos of *tea* in EgoPER. While each $a^l$ shows relatively stable and consistent action predictions, the results of $\hat{b}^l$ and $\hat{X}^l$ vary much more across videos. Therefore, this demonstrates how $\hat{b}^l$ provides more diverse action predictions than $a^l$.

**Ablation Study.** We explore the effectiveness of each module in DUBAD (Table 3), different $\beta$, $s$, and $l$ (Fig. 5 and 6). First, DUBAD achieves the best error detection performance on EgoPER, with F1@10 and F1@25 scores of 52.1% and 42.8%, respectively, outperforming other variants. Second, DUBAD achieves better sensitivity in $\hat{f}^s$ and $\hat{f}^l$ (27.6% on E-F1) while maintaining robustness in $f^s$ and $f^l$ (90.0% on C-F1) on EPIC-Tent-O compared to the variant in the top row in Table 3, which acts as a downgraded version of DUBAD. Furthermore, the middle row in Table 3 represents that the *robust* and *sensitive* action detectors share the same architecture but the weights are initialized

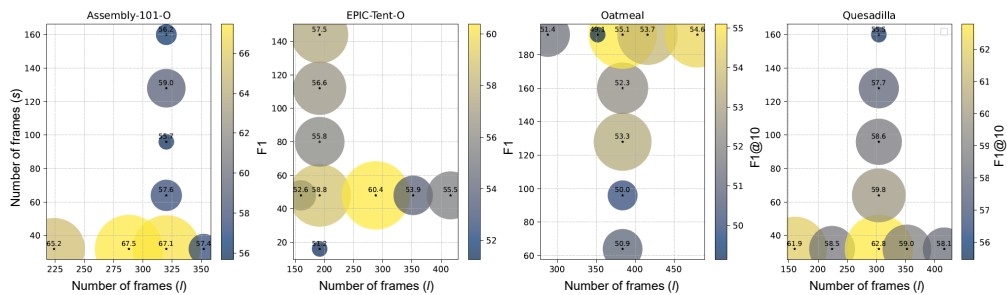

Figure 6: Performance with different $s$ and $l$ on Assembly-101-O, EPIC-Tent-O, and *oatmeal* and *quesadilla* in EgoPER. Lighter and larger circles indicate higher performance.

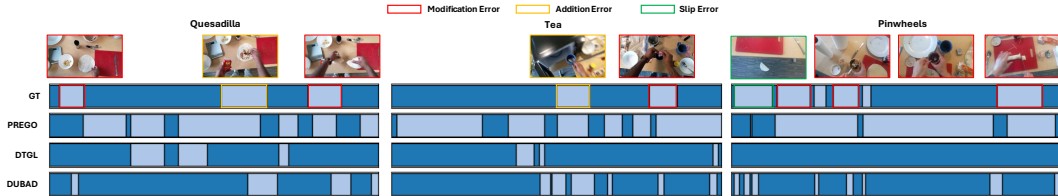

Figure 7: Qualitative visualization of online error detection on EgoPER. Each row from top to bottom shows specific erroneous frames, GT error detection, and error detection predicted by PREGO, DTGL, and DUBAD.

differently. The result shows that TAD improves the sensitivity of $\hat{f}^s$ and $\hat{f}^l$ to errors, increasing the E-F1 score from 19.5% (without TAD) to 27.6% on EPIC-Tent-O.

Next, Table 5 shows that the optimal margin differs over datasets, but overall $\beta = 0.6$ can yield decent performance for all datasets. In addition, we show the performance of different $s$ and $l$ using a bubble chart in Fig. 6 where y-axis and x-axis denote $s$ and $l$, and color and size represent the performance. The motivation of the design is based on the number of actions that $s$ and $l$ frames can see, causing different available semantic information along temporal dimension. See our supplementary material for detailed experiments and analysis.

**Qualitative Analysis.** We visualize the online error detection results generated by different models in Fig. 7. Specifically, PREGO produces many false positive segments and DTGL barely detects the locations of the errors. Our DUBAD can detect the locations of execution errors, including various types defined by EgoPER. For instance, DUBAD can detect (1) a modification error, "put tortilla on the table instead of the cutting board" (marked in red) in *quesadilla*, (2) an addition error, "put mug into microwave" (marked in yellow), and (3) an slip error, "drop tortilla on floor" (marked in green). See our supplementary material for more qualitative results.

## 5 CONCLUSION AND FUTURE WORK

In this paper, we propose DUal-Branch Action Detector (DUBAD) to perform *real-time and on-line error detection*. DUBAD consists of *robust* and *sensitive* action detectors trained with different temporal range to produce inconsistencies in predictions to detect procedural and execution errors. Our experiments on three datasets show that our framework efficiently detects complex errors more effectively in real-world videos than the baselines. On the other hand, we suggest several future directions for extending this work. First, an automatic solution for finding the optimal $s, l, \beta$ can make DUBAD more practical. Second, developing an architecture with explicit temporal information manipulation can provide better control over prediction inconsistency. Finally, transforming DUBAD into an online setting with a controlled delay (e.g., 3 seconds) can broaden its applicability to scenarios where higher accuracy is prioritized over real-time responsiveness.

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

# A  APPENDIX: SUPPLEMENTARY MATERIAL

## A.1  EXTENDED EXPERIMENTS AND ABLATIONS

In this section, we further explore DUBAD from different perspectives to better demonstrate its effectiveness and provide more experimental results.

**Action Prediction Analysis.** Table 4 shows the performance of using different action predictions for error detection. In particular, the first row from top of Table 4 demonstrates that $\hat{a}^s$ and $\hat{a}^l$ generated by *sensitive* action detectors using original frame features result in less inconsistency, achieving E-F1 scores of 60.4% and 9.1%, which are lower than DUBAD (74.1% and 27.6%) on Assembly-101-O and EPIC-Tent-O, respectively. In addition, the second and third rows show that robust action detectors using $b^s$ and $b^l$ sometimes results in unstable and inaccurate action predictions, leading to 0% on the C-F1 score on Assembly-101-O. Therefore, for *robust* predictions, we choose $a^s, a^l$ as they provide most stable and accurate predictions. For *sensitive* predictions, we choose $\hat{b}^s, \hat{b}^l$ as they demonstrate the best sensitivity to errors.

Table 4: Ablation study of different action predictions for error detection. Last row from top denotes DUBAD.

| Action Detectors | | EgoPER | | | Assembly-101-O | | | EPIC-Tent-O | | |
|---|---|---|---|---|---|---|---|---|---|---|
| *robust* | *sensitive* | F1@10 | F1@25 | F1@50 | Avg-F1 | E-F1 | C-F1 | Avg-F1 | E-F1 | C-F1 |
| $a^s, a^l$ | $\hat{a}^s, \hat{a}^l$ | 47.9 | 37.8 | 20.9 | 59.3 | 60.4 | 58.1 | 50.0 | 9.1 | **90.8** |
| $b^s, b^l$ | $\hat{a}^s, \hat{a}^l$ | 47.0 | 38.2 | 22.4 | 34.8 | 69.5 | 0.0 | 57.4 | 28.6 | 86.2 |
| $b^s, b^l$ | $\hat{b}^s, \hat{b}^l$ | 50.2 | 40.1 | 22.0 | 47.7 | **95.4** | 0.0 | **61.7** | **36.4** | 87.0 |
| $a^s, a^l$ | $\hat{b}^s, \hat{b}^l$ | **52.1** | **42.8** | **23.9** | **67.1** | 74.1 | 60.2 | 58.8 | 27.6 | 90.0 |

**Majority Voting Analysis.** Table 5 shows the performance with different numbers of mismatches. Note that one mismatch is considered as a loose constraint for capturing execution and procedural errors. The results show that the number of false positives for errors becomes higher when the numbers of mismatches are 2 or 1, resulting in lower C-F1 scores (2.9% and 50.0%) on both Assembly-101-O and EPIC-Tent-O. Therefore, we use a stricter threshold of three mismatches to better capture errors responsible for inconsistency across prediction pairs.

Table 5: Ablation on majority voting.

| Number of Mismatches | Assembly-101-O | | | EPIC-Tent-O | | |
|---|---|---|---|---|---|---|
| | Avg-F1 | E-F1 | C-F1 | Avg-F1 | E-F1 | C-F1 |
| 1 | 37.7 | 72.4 | 2.9 | 34.8 | 19.6 | 50.0 |
| 2 | 37.7 | 72.4 | 2.9 | 34.8 | 19.6 | 50.0 |
| 3 | **67.1** | **74.1** | **60.2** | **58.8** | **27.6** | **90.0** |

**Ablation Study of Loss.** Table 6 shows the performance with different losses. Specifically, $\mathcal{L}_{\text{TAD}}$ improves the sensitivity of sensitive action detectors, achieving the best results of 52.1% and 42.8% on F1@10 and F1@25, respectively, on EgoPER.

Table 6: Performance of applying different losses.

| Output | EgoPER | | | Assembly-101-O | | | EPIC-Tent-O | | |
|---|---|---|---|---|---|---|---|---|---|
| | F1@10 | F1@25 | F1@50 | Avg-F1 | E-F1 | C-F1 | Avg-F1 | E-F1 | C-F1 |
| $\mathcal{L}_{\text{CE}}^k + \mathcal{L}_{\text{TAD}}$ | 49.1 | 39.5 | **24.2** | 59.2 | 73.0 | 45.4 | 57.1 | 21.1 | **93.1** |
| $\mathcal{L}_{\text{CE}}^k + \mathcal{L}_{\text{Smo}}^k$ | 50.2 | 38.7 | 24.0 | 55.8 | 66.3 | 45.3 | 53.7 | 17.6 | 89.7 |
| $\mathcal{L}_{\text{CE}}^k + \mathcal{L}_{\text{Smo}}^k + \mathcal{L}_{\text{TAD}}^k$ | **52.1** | **42.8** | 23.9 | **67.1** | **74.1** | **60.2** | **58.8** | **27.6** | 90.0 |

**Action Detector Performance Analysis.** We explore the performance of our action detectors on EgoPER to show that DUBAD can capture errors while maintaining decent performance of temporal

Table 7: Performance of temporal action segmentation of different action predictions.

| Predictions | EgoPER | | | | | | | | | | | | | | | | | |
|---|---|---|---|---|---|---|---|---|---|---|---|---|---|---|---|---|---|---|
| | Quesadilla (F1@) | | | Oatmeal (F1@) | | | Pinwheel (F1@) | | | Coffee (F1@) | | | Tea (F1@) | | | All (F1@) | | |
| | 10 | 25 | 50 | 10 | 25 | 50 | 10 | 25 | 50 | 10 | 25 | 50 | 10 | 25 | 50 | 10 | 25 | 50 |
| $a^l$ | **74.1** | **69.2** | **50.8** | **80.7** | **75.4** | **60.3** | **59.2** | **49.4** | **36.3** | **68.0** | **58.6** | **37.0** | **62.4** | **54.2** | **39.7** | **68.9** | **61.4** | **44.8** |
| $a^s$ | 67.0 | 61.0 | 43.1 | 78.7 | 72.0 | 57.6 | 57.3 | 46.9 | 29.1 | 65.2 | 55.5 | 33.7 | 59.3 | 48.5 | 31.7 | 65.5 | 56.8 | 39.0 |
| $\hat{b}^l$ | 69.1 | 62.4 | 42.9 | 75.4 | 70.0 | 50.7 | 53.5 | 46.7 | 30.6 | 63.2 | 53.2 | 32.0 | 60.3 | 53.1 | 37.1 | 64.3 | 57.1 | 38.7 |
| $\hat{b}^s$ | 66.7 | 60.9 | 43.2 | 73.4 | 66.1 | 45.2 | 54.1 | 44.7 | 28.3 | 62.5 | 54.2 | 32.1 | 58.7 | 50.4 | 33.3 | 63.1 | 55.3 | 36.4 |

Table 8: Detailed performance of error detection.

| Method | Assembly-101-O | | | | | | |
|---|---|---|---|---|---|---|---|
| | Avg-F1 | E-F1 | E-Precision | E-Recall | C-F1 | C-Precision | C. Recall |
| PREGO | 32.5 | 41.8 | 27.8 | **84.1** | 23.1 | 68.8 | 13.9 |
| DTGL | 53.5 | 28.1 | 22.5 | 37.3 | **78.9** | **85.0** | 73.5 |
| DUBAD | **67.1** | **74.1** | **64.8** | 56.9 | 60.2 | 71.4 | **76.9** |

| Method | EPIC-Tent-O | | | | | | |
|---|---|---|---|---|---|---|---|
| | Avg-F1 | E-F1 | E-Precision | E-Recall | C-F1 | C-Precision | C. Recall |
| PREGO | 29.4 | 17.2 | 9.5 | **93.3** | 41.6 | **97.9** | 26.4 |
| DTGL | 46.5 | 23.7 | **73.3** | 14.1 | 69.3 | 54.4 | **95.2** |
| DUBAD | **58.8** | **27.6** | 28.6 | 26.7 | **90.0** | 89.6 | 90.5 |

action segmentation. We follow the same metric as in Farha & Gall (2019) which evaluates the performance of action predictions regarding classification and localization. Note that we ignore the erroneous frames and only evaluate on normal frames. The main reason is that certain errors in EgoPER (e.g., addition errors and some slip errors) neither belong to nor resemble any seen actions, and thus should not be included when evaluating temporal action segmentation. Table 7 reports the F1 scores of four action predictions from our *robust* and *sensitive* action detectors. In particular, the action prediction $a^l$ produced by the *robust* action detector $f^l$ yields the highest F1 scores, achieving 68.9%, 61.4%, and 44.8% on F1@10, F1@25, and F1@50, respectively, demonstrating strong capabilities on temporal action segmentation with SAM. On the other hand, although the action prediction $\hat{b}^l$ generated by the *sensitive* action detectors performs worse than $a^l$, it achieves competitive results, with F1 scores of 64.3%, 57.1%, and 38.7% on F1@10, F1@25, and F1@50.

**Detailed Experimental Results.** We conduct further experiments to validate our method. Table 8 shows the detailed error detection performance of the baselines and DUBAD on Assembly-101-O and EPIC-Tent-O. Our method consistently has good performance on precision and recall, specifically achieving 64.8% on E-Precision and 89.6% on C-Precision compared to 22.5% and 54.4% by DTGL on Assembly-101-O and EPIC-Tent-O, respectively. In addition, we report the counts of true positives, false positives, false negatives, and true negatives for the erroneous actions in Table 9. Note that each sample represents an predicted action. The results show that our method does not over-segment nor under-segment the videos, while achieving good performance on both datasets.

Next, we provide more heatmap visualizations for $\hat{X}^l$ on errors in Fig. 8. Specifically, the right column in Fig 8 (b) shows that the TAD module generates inconsistent features for errors and therefore, produce inconsistency between $a^l$ and $\hat{b}^l$. However, the TAD module also produces inconsistent features for correct actions, resulting in wrong action predictions (see the left column in Fig. 8 (b)). As a result, there is still a room for improving both *robust* and *sensitive* action detectors.

Finally, Table 10 and 11 shows the ablation study of *robust* and *sensitive* action detectors on Assembly-101-O, EPIC-Tent-O, and EgoPER. Note that higher overlap thresholds (e.g., F1@50) impose stricter localization requirements, whereas lower thresholds primarily reflect detection performance. Although DUBAD (the last row of Table 11) achieves a 0.3% lower F1@50 than the branch without the TAD module, it improves F1@10 and F1@25 by 4.4% and 4.7%, respectively, demonstrating better error detection capability with comparable localization performance.

| Dataset | TP | FP | FN | TN |
|---|---|---|---|---|
| Assembly-101-O | 140 | 56 | 42 | 74 |
| EPIC-Tent-O | 4 | 10 | 11 | 95 |

Table 9: The number of true positives (TP), false positives (FP), false negatives (FN), and true negatives (TN) regarding error actions generated by DUBAD on Assembly-101-O and EPIC-Tent-O.

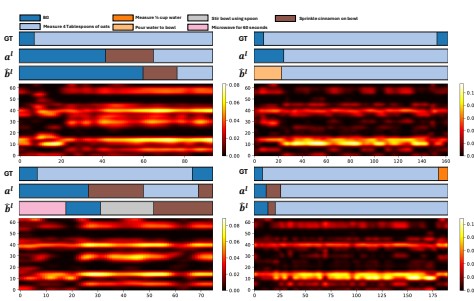 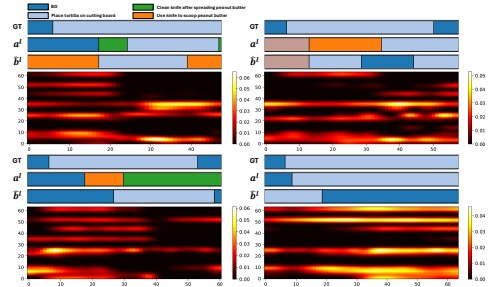

(a) Left column represents two correct actions. Right column represents two erroneous actions: *directly pour oats into bowl without measuring*.

(b) Left column represents two correct actions. Right column represents two erroneous actions: *drop tortilla on the floor*.

Figure 8: Each row in a subfigure, from top to bottom shows frame-wise ground-truth (GT) actions, $a^l$, and $\hat{b}^l$, as well as the heatmap visualization of $\hat{x}^l$ for errors on *oatmeal* (a) and *pinwheels* (b) of EgoPER. For $\hat{x}^l$, x-axis represents the number of channels (we only visualize 64 of them) and y-axis denotes the number of frames.

## A.2 MULTI-HEAD ATTENTION ARCHITECTURE

We describe the details of the Multi-Head Attention module MHA(Q, K, V) in Section 3.2 formulated as

$$\text{MHA}(Q, K, V) = \text{Concat}(head_1, \dots, head_u)W^O, \tag{4}$$

$$\text{where } head_i = \text{Attention}(QW_i^Q, KW_i^K, VW_i^V), \tag{5}$$

$$\text{Attention}(Q, K, V) = \text{softmax}\left(\frac{QK^\top}{\sqrt{d}}\right)V. \tag{6}$$

where $Q$ is a frame feature $x$, $K$ and $V$ are the action prototype $h$, $W^O, W_i^Q, W_i^K, W_i^V$ are learnable weights, $u$ is the number of heads, and $d = \frac{D}{u}$. The MHA module aims to reconstruct the frame feature by weighted sum over the similarities between the action prototypes and the current frame feature.

## A.3 HYPERPARAMETER COMBINATION SETTING

Table 12 shows the optimal hyperparameter setting of DUBAD for each dataset. In this section, for each hyperparameter, we provide further experiments and analysis.

**Supplementary Experiments.** We provide the experiments for $s$, $l$ frames and $\beta$ on other tasks of EgoPER. Figure 9 shows the performance of different $s$ and $l$ frames on *tea*, *pinwheels*, and *coffee* in EgoPER. Since the numbers of frames for $s$ and $l$ vary across datasets, developing an automatic solution to make the framework more practical is crucial for future work. We therefore provide some preliminary ideas and analysis in the following section. Figure 10 shows the performance of different margins $\beta$ on *quesadilla*, *oatmeal*, and *pinwheels* of EgoPER. Same as in the main paper, $\beta = 0.6$ can yield decent performance for all datasets.

**Causal Mode Filter Analysis.** Figure 11 shows the performance of various window sizes for the causal mode filter on three datasets. The size of mode filter strongly influences the performance of error detection on the dataset with shorter average action duration such as EgoPER. Specifically, the

Table 10: Ablation study of DUBAD. RGB refers to the pipeline shown by the solid line in Fig. 1.

| $f^s$ and $f^l$ | | $\hat{f}^s$ and $\hat{f}^l$ | | | Assembly-101-O | | | EPIC-Tent-O | | |
| RGB | SAM | RGB | SAM | TAD | Avg-F1 | E-F1 | C-F1 | Avg-F1 | E-F1 | C-F1 |
|---|---|---|---|---|---|---|---|---|---|---|
| ✓ | | ✓ | ✓ | | 52.8 | 22.1 | 83.4 | 53.7 | 17.1 | 90.2 |
| | ✓ | ✓ | ✓ | ✓ | 47.1 | 32.6 | 61.5 | 56.5 | 30.3 | 82.7 |
| ✓ | | ✓ | ✓ | ✓ | 45.1 | 22.3 | 67.9 | 53.4 | 14.8 | 92.0 |
| ✓ | ✓ | | ✓ | ✓ | 60.8 | 71.8 | 49.8 | 54.2 | 19.5 | 88.9 |
| ✓ | ✓ | ✓ | | ✓ | 59.3 | 76.7 | 42.0 | 53.7 | 16.0 | 91.4 |
| ✓ | ✓ | ✓ | ✓ | | 59.6 | 71.0 | 48.2 | 53.5 | 19.5 | 87.5 |
| ✓ | ✓ | ✓ | ✓ | ✓ | **67.1** | 74.1 | 60.2 | **58.8** | 27.6 | 90.0 |

Table 11: Ablation study of DUBAD on EgoPER. RGB denotes the solid line in Fig. 1.

| $f^s$ and $f^l$ | | $\hat{f}^s$ and $\hat{f}^l$ | | | EgoPER | | |
| RGB | SAM | RGB | SAM | TAD | F1@10 | F1@25 | F1@50 |
|---|---|---|---|---|---|---|---|
| ✓ | | ✓ | ✓ | | 47.8 | 37.6 | 22.6 |
| | ✓ | ✓ | ✓ | ✓ | 47.1 | 37.1 | 20.1 |
| ✓ | | ✓ | ✓ | ✓ | 47.8 | 37.8 | 20.9 |
| ✓ | ✓ | | ✓ | ✓ | 48.6 | 38.8 | 21.7 |
| ✓ | ✓ | ✓ | | ✓ | 49.9 | 39.6 | 23.0 |
| ✓ | ✓ | ✓ | ✓ | | 47.7 | 38.1 | 24.2 |
| ✓ | ✓ | ✓ | ✓ | ✓ | 52.1 | 42.8 | 23.9 |

performance drops from $36.9\%, 55.1\%, 46.6\%, 62.8\%$, and $56.8\%$ to $25.2\%, 39.7\%, 29.6\%, 44.2\%$, and $36.5\%$ when the window size $m$ decreases from 30 to 10. This indicates that 1) the action predictions need to be smoothed though smoothing loss is applied, and 2) the smoothing does not annihilate inconsistencies in predictions.

**Semantic of Temporal Information Analysis.** We investigate the semantic difference in terms of temporal information. Figure 12, 13, 14, and 15 show the numbers of actions appearing in different numbers of frames ($s$ and $l$) of a sliding window on three datasets. In practice and our main paper, we intend to observe one to two actions with $s$ frames and three to four actions with $l$ frames, respectively. Specifically, for Assembly-101-O and EPIC-Tent-O (Fig. 12), $s = 32$ and $s = 48$ can almost only identify one action, but $l = 320$ and $l = 192$ often identify more than two actions in the sliding window. On the other hand, for *tea* and *quesadilla* of EgoPER (Fig. 13), $s = 32$ and $s = 56$ can almost only identify one action, but $l = 304$ and $l = 416$ often identify more than three or four actions in the sliding window. The semantic difference regarding number of actions influences the prediction behavior especially when distribution shift occurs (i.e., procedural errors occur). Note that the average action duration in Assembly-101-O and EPIC-Tent-O is longer than EgoPER.

## A.4 Detailed Qualitative Visualization

In this section, we provide detailed visualization of action predictions and error detection on EgoPER (see Fig. 16, 17, 18, 19, and 20). In each figure, each row from top to bottom represents RGB frames showing the errors, GT actions, $a^s, \hat{b}^s, a^l, \hat{b}^l$, GT errors, and predicted errors by DUBAD. In particular, in *tea* of EgoPER (Fig. 16), the first error "directly pour water into kettle without measuring" in the left figure shows that DUBAD produce the inconsistency in $a^l$ and $\hat{b}^l$. In addition, the second error "put mug into microwave for 5 seconds" reveals inconsistencies not only between *robust* and *sensitive* action predictions but also in temporal differences ($a^s$ vs. $a^l$). Let us see another example in *quesadilla* of EgoPER (Fig. 17). There are three different action predictions among $a^s, a^l, \hat{b}^s$, and $\hat{b}^l$ for the third error "tear tortilla into two pieces with hands instead of using knife" in the left figure of Fig. 17. Overall, the visualizations of action predictions demonstrate that our DUBAD produces inconsistencies in predictions when errors occur, facilitating error detection.

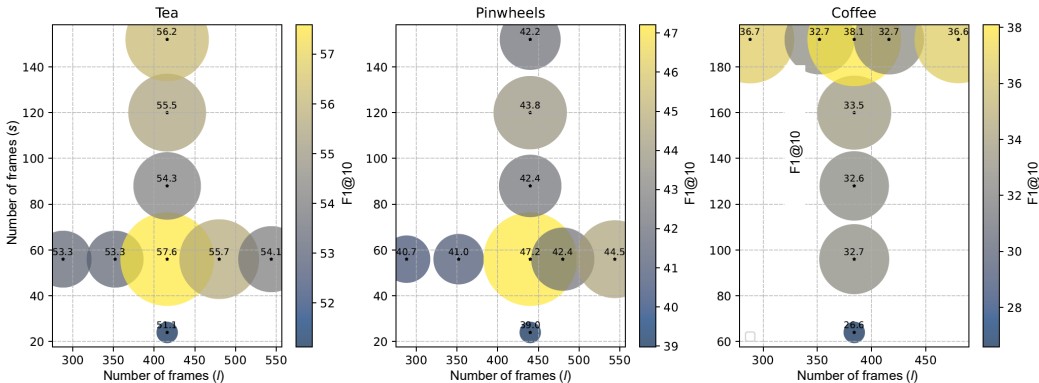

Figure 9: Performance with different $s$ and $l$ on *tea*, *pinwheels*, and *coffee* in EgoPER.

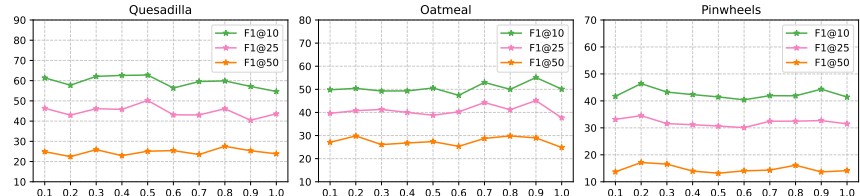

Figure 10: Performance of different margins $\beta$ on *quesadilla*, *oatmeal*, and *pinwheels* of EgoPER.

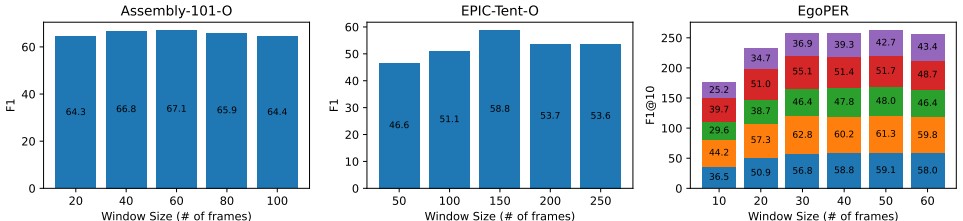

Figure 11: Performance of different sizes of causal mode filters. For the right-most figure, purple, red, green, orange, and blue color denote *coffee*, *oatmeal*, *pinwheels*, *quesadilla*, and *tea* of EgoPER, respectively.

Table 12: Optimal combination of hyperparameters $s$, $l$, $\beta$, and $m$ for DUBAD.

| Hyperparameter | EgoPER | | | | | Assembly-101-O | Epic-Tent-O |
| | Quesadilla | Tea | Oatmeal | Pinwheels | Coffee | | |
|---|---|---|---|---|---|---|---|
| $s$ (number of frames) | 32 | 56 | 192 | 56 | 192 | 32 | 48 |
| $l$ (number of frames) | 304 | 416 | 384 | 440 | 384 | 320 | 192 |
| $\beta$ (TAD margin) | 0.5 | 0.6 | 0.9 | 0.2 | 0.8 | 0.2 | 0.6 |
| $m$ (causal mode filter) | 30 | 30 | 30 | 30 | 30 | 60 | 150 |

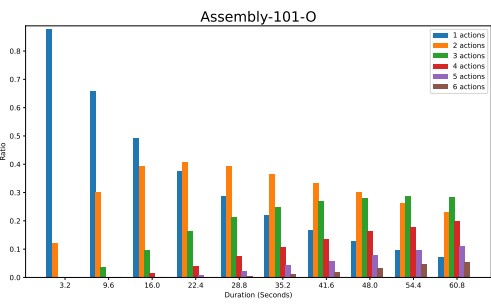

(a) The numbers of actions regarding different of frames on Assembly-101-O.

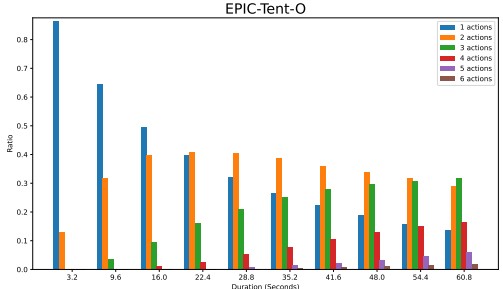

(b) The numbers of actions regarding different of frames on EPIC-Tent-O.

Figure 12: The analysis for the numbers of actions regarding the numbers of frames of a sliding window.

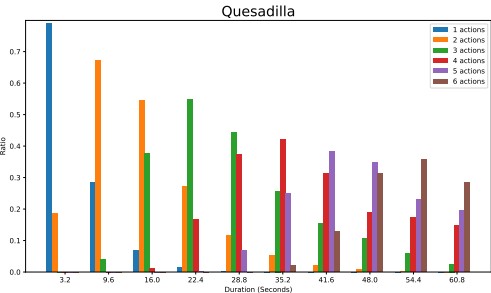

(a) The numbers of actions regarding different of frames on *quesadilla* of EgoPER.

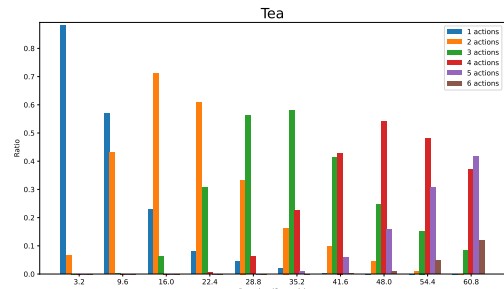

(b) The numbers of actions regarding different of frames on *tea* of EgoPER.

Figure 13: The analysis for the numbers of actions regarding the numbers of frames of a sliding window.

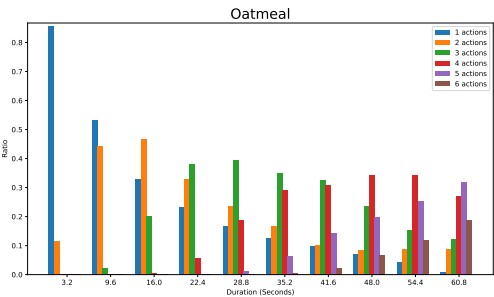

(a) The numbers of actions regarding different of frames on *oatmeal* of EgoPER.

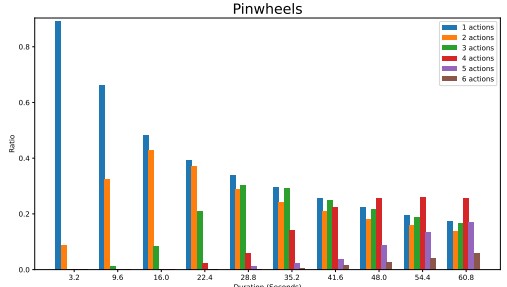

(b) The numbers of actions regarding different of frames on *pinwheels* of EgoPER.

Figure 14: The analysis for the numbers of actions regarding the numbers of frames of a sliding window.

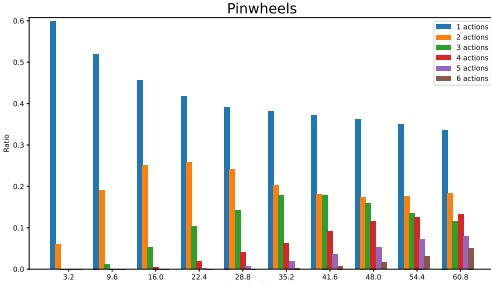

Figure 15: The numbers of actions regarding different of frames on *coffee* of EgoPER.

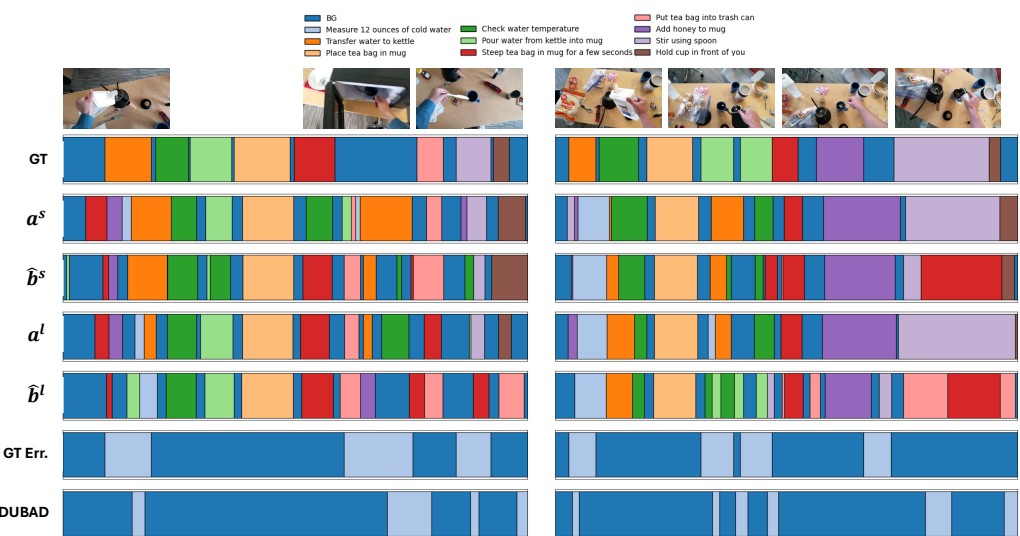

Figure 16: Qualitative results of our proposed method on *tea* of EgoPER. Error descriptions for the left figure: "directly pour water into kettle without measuring", "addition: put mug into microwave for 5 seconds", "stir mug with knife instead of spoon". Error descriptions for the right figures: "directly pour water into kettle without measuring", "pour water into a wrong mug", "correction: put tea bag into the mug with water", "sprinkle cinnamon into mug".

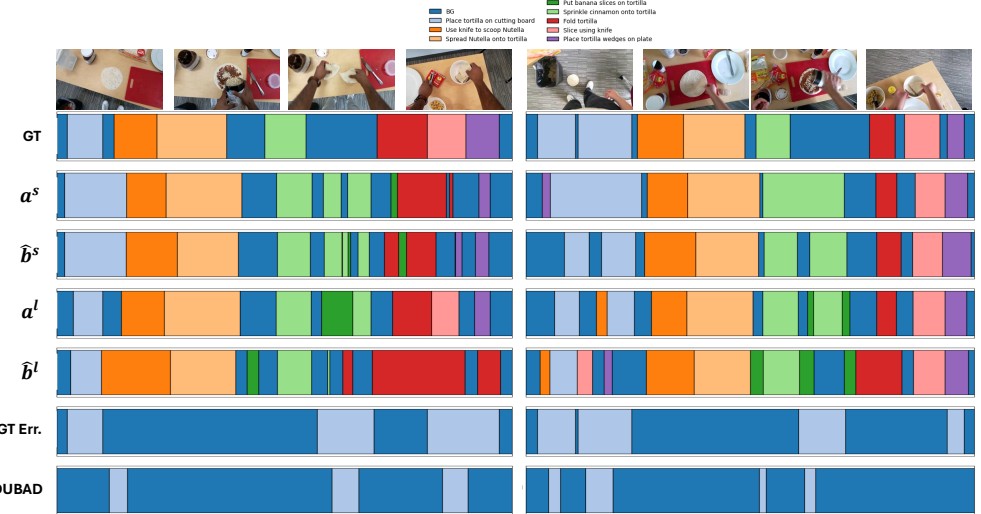

Figure 17: Qualitative results of our proposed method on *quesadilla* of EgoPER. Error descriptions for the left figure: "put tortilla on table instead of cutting board", "addition: pour oats onto tortilla", "tear tortilla into two pieces with hands instead of using knife", "put tortilla wedges into bowl in stead of plate". Error descriptions for the right figure: "drop tortilla on floor", "correction: put a new tortilla on cutting board", "addition: pour oats onto tortilla", "put tortilla wedges into bowl instead of plate".

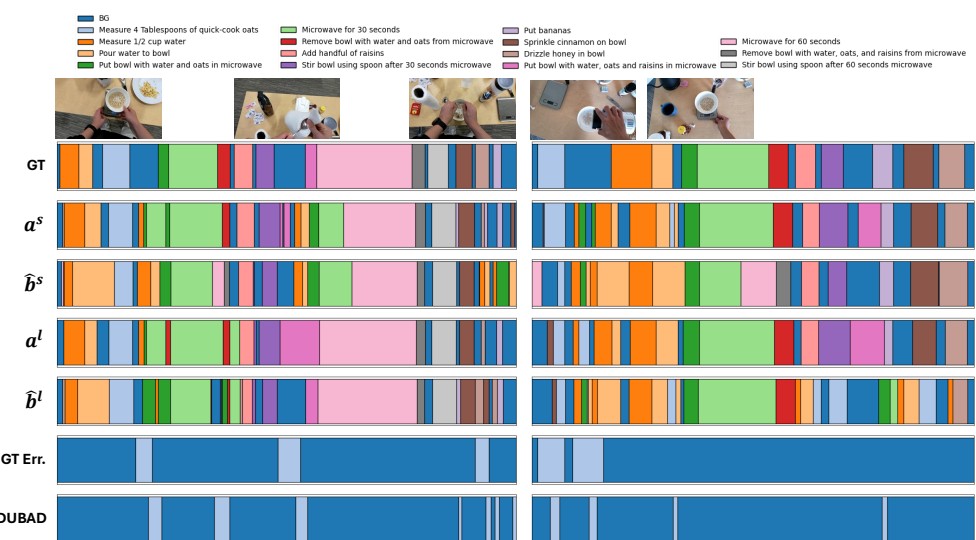

Figure 18: Qualitative results of our proposed method on *oatmeal* of EgoPER. Error descriptions for the left figure: "addition: put bowl with oats on a weighing scale", "addition: clean spoon after stirring", "add sugar into bowl instead of honey". Error descriptions for the right figure: "directly pour oats into bowl without measuring", "addition: put bowl with oats on a weighing scale".

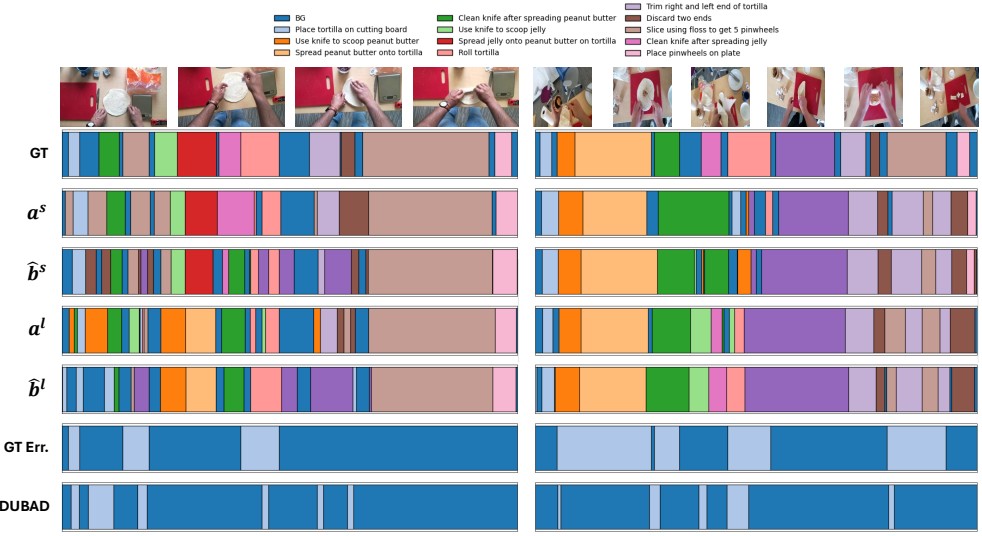

Figure 19: Qualitative results of our proposed method on *pinwheels* of EgoPER. Error descriptions for the left figure: "place tortilla on table instead of cutting board", "addition: add sugar onto tortilla", "fold tortilla into half", "correction: unfold tortilla and roll it". Error descriptions for the right figure: "scoop peanut butter using spoon instead of knife", "spread peanut butter onto tortilla using spoon instead of knife", "clean spoon instead of knife", "fold tortilla into half", "correction: unfold tortilla and roll it", "slice tortilla to get pinwheels using knife instead of knife".

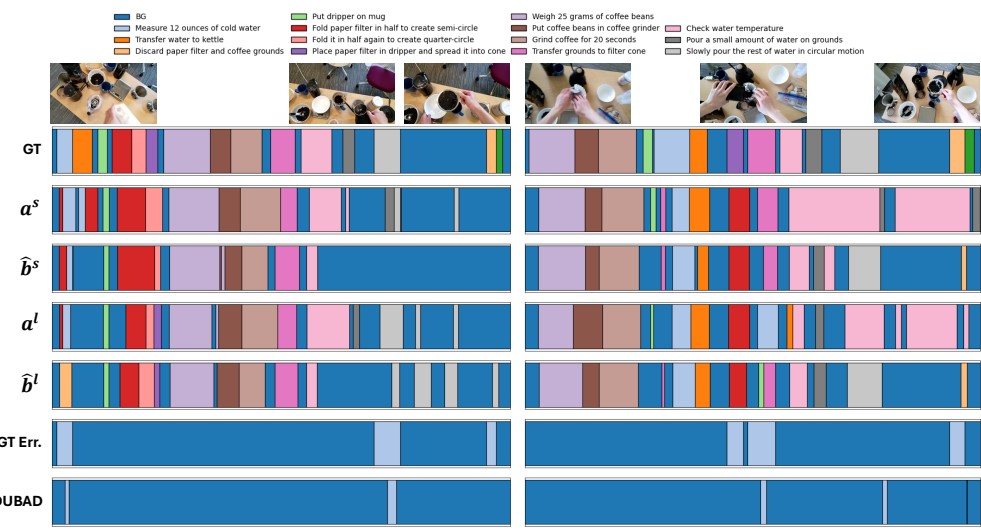

Figure 20: Qualitative results of our proposed method on *coffee* of EgoPER. Error descriptions of the left figure: "measure incorrect amount of water", "pour water from kettle into filter cone without circular motion", "discard paper filter before drained". Error descriptions for the right figure: "squeeze paper filter into cone without folding", "spill out coffee grounds while transferring coffee grounds to grinder", "discard paper filter before drained".

