# OpenReview forum: "Leveraging Prediction Inconsistency for Online Error Detection in Procedural Videos"
_ICLR.cc/2026/Conference — Submitted to ICLR 2026_

### Official Review · Reviewer_79aK · 2025-10-29

**Soundness:** 2
**Presentation:** 2
**Contribution:** 2
**Rating:** 4
**Confidence:** 4

**Summary:**

This paper introduced DUal-Branch Action Detector (DUBAD) for online error detection in procedural videos. The system is designed to work with egocentric videos from wearable devices and can detect errors as they occur. DUBAD combines a robust action detector and a sensitive action detector. The robust action detector generates accurate action prediction and the sensitive action detector generates inconsistent predictions when errors occur. The proposed method was evaluated on EgoPER, Assembly-101-O and EPIC-Tent-O. The state-of-the-art performance demonstrates its effectiveness.

**Strengths:**

1. The proposed DUal-Branch Action Detector (DUBAD) combines robust and sensitive action detectors, trained with differing temporal receptive fields, to capture distinct error types. This dual design is an original contribution in the online error detection domain.
2. The method conceptualizes prediction disagreement as an implicit measure of uncertainty, which is an innovative way of bridging temporal modeling and online reliability estimation.
3. This paper introduces two modules — the Step Attention Module (SAM) for robust detection and the Temporal-Aware Dynamic (TAD) module for error-sensitive adaptation — represents a meaningful architectural advancement that extends beyond a simple ensemble strategy.

**Weaknesses:**

1. Training only on error-free videos (one-class classification) limits what the model can learn. The method requires dataset-specific tuning of multiple hyperparameters (s, l, β, m)
2. While achieving 24.4 FPS, this is slower than DTGL (37.0 FPS) despite similar lightweight goals. And feature extraction time (0.0261s) not counted as part of their method but is required for deployment
3. Limited explanation for why dual branches with different temporal receptive fields is the optimal design
4. Related work part is poorly written, missing majority of work in the field such as:
[1] Xu, Mingze, et al. "Temporal recurrent networks for online action detection." Proceedings of the IEEE/CVF international conference on computer vision. 2019.
[2] Xu, Mingze, et al. "Long short-term transformer for online action detection." Advances in Neural Information Processing Systems 34 (2021): 1086-1099.
[3] Guo, Hongji, et al. "Uncertainty-based spatial-temporal attention for online action detection." European Conference on Computer Vision. Cham: Springer Nature Switzerland, 2022.
etc.

**Questions:**

Different people can perform procedural action in different order, how to handle this when detection errors?

---

> ### Author Response · Authors · 2025-11-19
> **Reviewer 79aK Rebuttal**
>
> We sincerely thank the reviewer for the comments.
>
> - **One-class Training and Hyperparameter Limitations**: Our work follows the same problem setting as prior error detection studies (a one-class classification scenario in which only normal videos are available during training). This setting mitigates the cost and difficulty of collecting diverse erroneous videos and their annotations, which are often expensive and time-consuming to obtain. Moreover, existing datasets do not provide sufficient or diverse examples of erroneous executions for effective supervised training.
>
> - **Inference Speed Issue**: Although our method is slower than DTGL, it still operates in real time. In addition, our approach outperforms DTGL on error detection across EgoPER, Assembly-101-O, and Epic-Tent-O (Table 1), demonstrating its advantages in both efficiency and effectiveness.
>
> - **Limited Explanation for Model Design**: Our dual-branch method trained different temporal receptive fields aims to generate inconsistencies in predictions when execution and procedural errors occur in videos. Specifically, inconsistencies between sensitive and robust predictions highlight spatiotemporal differences that may reveal execution errors (e.g., placing a tortilla on a cutting board versus a table) as discussed in lines 266–269. On the other hand, to capture potential procedural errors (e.g., missing spread peanut butter before spread jelly), we focus on inconsistencies between detectors that rely on different amounts of past temporal
> context and therefore respond differently when an action deviates from the expected sequence, as mentioned in lines 293–296.
>
> - **Poorly Written Related Work**: We thank the reviewer for the comment. We will update our related work with suggested references accordingly.
>
> - **Action Variations in a Procedure**: We follow the same problem setting as prior work, described in One-class Training and Hyperparameter Limitations. We assume that the training videos cover all valid execution sequences of actions consistent with a task graph or recipe. In this setting, variations of an action are treated as errors, following the definition used in EgoPER. For example, the action “stir mug using spoon” may appear in the training data, while its variation “stir mug using knife”, although still capable of completing the task, is categorized as a modification error in EgoPER.

---

### Official Review · Reviewer_Psmx · 2025-10-31

**Soundness:** 3
**Presentation:** 4
**Contribution:** 3
**Rating:** 6
**Confidence:** 2

**Summary:**

This paper introduces DUBAD for online mistake detection in procedural videos. DUBAD addresses the limitations of existing methods that they cannot detect multiple errors in a video, or only focus on procedural errors while missing execution errors.

**Strengths:**

**Originality**:
- This paper is based on a clear motivation: online mistake detection needs to detect two types of errors and to perform detection continuously, rather than stopping after the first error is detected.

**Quality**:
- This paper is complete, from motivation to methodology to experiments.

**Clarity**:
- This paper clearly elaborates the limitations of existing methods and, based on this, proposes a targeted solution.

**Significance**:
- This paper addresses an important issue in online mistake detection and the experiments show that the proposed method achieves superior results.

**Weaknesses:**

- The most recent method used for comparison in the experiments was published in 2024. Is it possible to include newer methods in the comparison?
- I think one important reference [1] has been omitted, which also focuses on Procedural and Execution Mistakes. I hope the authors would discuss the difference of this paper's motivation from [1].

[1] Patsch, Constantin, et al. "MistSense: Versatile Online Detection of Procedural and Execution Mistakes." Proceedings of the IEEE/CVF International Conference on Computer Vision. 2025.

**Questions:**

- Given that many LLM-based models for online video understanding already exist (e.g., Videollm-online[2]), is it also possible to achieve real-time detection for LLM-based online mistake detection? If so, what is the necessity of continuing to use small models? If not, could the authors provide experimental data demonstrating that LLM-based methods have significantly higher latency than non-LLM methods?

[2] Chen, Joya, et al. "Videollm-online: Online video large language model for streaming video." Proceedings of the IEEE/CVF Conference on Computer Vision and Pattern Recognition. 2024.

---

> ### Author Response · Authors · 2025-11-19
> **Reviewer Psmx Rebuttal**
>
> We sincerely thank the reviewer for the comments and for recognizing the originality, quality, clarity, and significance of our work.
>
> - **Latest Baseline**: Yes, thank you for the suggestion. We compare our method with MistSense [1] on EPIC-Tent-O (see table below) and will include this comparison in the paper. The performance of MistSense is taken directly from its original paper. Our method achieves a comparable Avg-F1 (58.8\%) to MistSense (59.8\%) on EPIC-Tent-O. However, MistSense requires a higher computational cost. As reported in Section 4.6 of MistSense, it requires roughly 4 seconds for error detection and explanation on the same RTX A6000 GPU. Under the same evaluation setup for error detection, our method requires approximately 75 GFLOPs (feature extractor + action detectors), which is lower than the ~91.6 GFLOPs needed by MistSense (ViT-L + two Q-Formers).
>
> |Method|Avg-F1|E-F1|C-F1|
> |---|---|---|---|
> |MSGI | 44.5 | 22.0 |  66.9 |
> |$\textrm{MSG}^2$|45.2 | 22.9 | 67.5|
> |PREGO | 29.4 | 17.2 | 41.6|
> |DTGL | 46.5 | 23.7 | 69.3|
> |MistSense |**59.8** | **89.7** | 29.8|
> |DUBAD (Ours) | 58.8 | 27.6 | **90.0**|
>
> - **Missing Reference**: We thank the reviewer for the suggestion. MistSense utilizes both RGB frames and hand poses mainly improves error detection and incorporates error explanation for detected errors using large language models without considering real-time processing. Our method considers both performance and real-time efficiency for online error detection.
>
> - **Real-time LLM-based Online Mistake Detection**: Due to the limited hardware capabilities of wearable devices such as the Microsoft HoloLens and Apple Vision Pro, lightweight models remain essential, particularly for running applications like error detection directly on these devices (lines 42-45). Meanwhile, the PREGO baseline in Table 1 is an LLM-based method that operates at only about 1 FPS (see Table 2), which is far below the real-time requirement even when executed on a high-end GPU (NVIDIA RTX A6000).

---

> > ### Comment · Reviewer_Psmx · 2025-11-19
> >
> > Thanks for the authors' response, but I still have some questions as follows:
> > Could the author discuss more about the weird comparative results between DUBAD and MistSense, where the E-F1 and C-F1 scores show totally different trends?
> > Also, compared with MistSense, I think that only taking real-time efficiency into consideration is not novel enough. I hope the author can further clarify the novelty of this paper beyond MistSense.

---

> > > ### Author Response · Authors · 2025-11-19
> > > **Reviewer Psmx Rebuttal 2**
> > >
> > > Thank you for the reviewer’s prompt response.
> > >
> > > In summary, MistSense tends to overly predict errors (higher E-F1), whereas our method tends to classify actions as correct (higher C-F1). This behavior may result from the fact that MistSense is trained on both normal and erroneous videos and is therefore more sensitive to the class imbalance, while our method is trained only on normal videos. Our method exhibits prediction behavior more similar to PREGO and DTGL, while achieving higher accuracy in detecting actual errors.
> > >
> > > Below is a summary of the novelty of our work beyond MistSense:
> > >
> > > - **Training paradigm**: MistSense is trained in a fully supervised setting (see Section 3.3 of MistSense) using both normal and erroneous videos. In contrast, our method follows the one-class problem setting used in prior work (PREGO and DTGL) and is trained only on normal videos, while still achieving comparable performance. This makes our approach more practical for real-world scenarios where collecting diverse erroneous samples is costly and challenging.
> > >
> > > - **Efficiency**: Computational efficiency is a key focus of our work, as it is crucial for deploying online error detection systems on wearable devices that lack high-end hardware. Our method is more efficient than MistSense, making it better suited for such platforms with constrained resource.

---

> > > > ### Comment · Reviewer_Psmx · 2025-11-25
> > > >
> > > > The authors have addressed my concerns.
> > > >
> > > > I decide to maintain my positive rating of 6.

---

### Official Review · Reviewer_c1rb · 2025-10-31

**Soundness:** 2
**Presentation:** 1
**Contribution:** 2
**Rating:** 2
**Confidence:** 4

**Summary:**

The paper proposes DUBAD (DUal-Branch Action Detector), a framework for real-time online error detection in procedural videos. The key idea is to exploit prediction inconsistencies between robust and sensitive action detectors to identify both execution and procedural errors. The model combines two complementary modules — the Step Attention Module (SAM) for robustness and the Temporal-Aware Dynamic (TAD) module for sensitivity — each trained with different temporal receptive fields. The authors evaluate DUBAD on EgoPER, Assembly-101-O, and EPIC-Tent-O, showing that it achieves state-of-the-art F1 scores with real-time inference speed.

**Strengths:**

The proposed method achieves strong quantitative results across multiple benchmarks, outperforming prior online error detection methods such as PREGO and DTGL.

**Weaknesses:**

- Figure 1 is visually unclear (Too small texts!) and unintuitive; it is hard to follow how the different modules (SAM, TAD, robust/sensitive detectors) interact. The graphical elements are cluttered and do not effectively convey the main conceptual flow.

- The notation is overly complex and not well presented. For example, lowercase h denotes action prototypes, uppercase H denotes concatenated weight–bias vectors, and k appears both above the sigma operator and as a subscript to H in Equation (3), which creates confusion.

- The design philosophy of SAM and TAD is not clearly motivated or justified. They appear somewhat arbitrary, and it is difficult to grasp a coherent conceptual rationale for how these modules interact to produce “prediction inconsistency.”

- Table 3 (Ablation Study) uses a simple checkbox-style presentation that does not effectively demonstrate the causal contribution of each component. It is unclear whether the performance gains genuinely reflect the intended functionality of SAM and TAD or are due to confounding factors in training.

- Despite good performance numbers, the overall framework feels overly complicated, with numerous loss terms and modules that obscure interpretability. The complexity of the design risks overshadowing the conceptual clarity of the main contribution.

**Questions:**

Please see weaknesses.

---

> ### Author Response · Authors · 2025-11-19
> **Reviewer c1rb Rebuttal**
>
> We sincerely appreciate the reviewer’s valuable and constructive comments.
>
> - **Unclear Figure**: Thank you for your comments. We will update the figure accordingly to address your concern.
>
> - **Overly Complex Notation**: Thank you for your comments. We will simplify and update the notations to avoid confusion.
>
> - **Unclear Design Philosophy of SAM and TAD**: Our SAM module uses cross attention to distill input features with action prototypes, which capture the general semantics of their corresponding actions, enabling the model to generate robust and generalizable representations. To encourage differentiated predictions when potential execution errors occur, our TAD module injects error sensitivity into SAM’s features by applying input-dependent dynamic weights that are regularized to remain sparse.
>
> - **Ineffective Ablation Study**: Table 3, 10, and 11 report the performance of each proposed component based on the fixed hyperparameters in Table 12 for each dataset. We will update the paper to clarify the concern.
>
> - **Ovearly Complicated Framework**: The core idea of our framework is to detect execution and procedural errors in videos by identifying inconsistencies in model predictions. To capture execution errors, we detect inconsistencies between the robust and sensitive action detectors when spatiotemporal differences occur (e.g., using a spoon instead of a knife or accidentally dropping the spoon), making the action deviate from the normal one. To capture procedural errors, we detect inconsistencies between detectors that rely on different amounts of past temporal context and therefore respond differently when an action deviates from the expected sequence. Based on this core idea, our framework integrates SAM and TAD and trains them with different temporal window sizes, enabling the model to naturally produce the prediction inconsistencies required for detecting both types of errors.

---

### Official Review · Reviewer_z56d · 2025-10-31

**Soundness:** 2
**Presentation:** 3
**Contribution:** 2
**Rating:** 2
**Confidence:** 4

**Summary:**

The paper proposes DUBAD, a dual-branch online error-detection framework for procedural videos that explicitly leverages prediction inconsistency between a robust and a sensitive action detector, each trained with different temporal receptive fields. The robust branch uses a Step Attention Module (SAM) conditioned on action prototypes, while the sensitive branch adds a Temporal-Aware Dynamic (TAD) module to induce input-dependent affine transformations that amplify inconsistencies under errors. The system flags errors when ≥3 of 4 pairwise comparisons disagree. Experiments on EgoPER, Assembly-101-O, and EPIC-Tent-O show state-of-the-art F1 and real-time throughput with a lightweight architecture.

**Strengths:**

1.	The proposed DUBAD framework demonstrates good performance improvements over state-of-the-art baselines across three challenging and diverse datasets, validating the effectiveness of the proposed approach.
2.	The method is designed to be lightweight and efficient, achieving real-time processing speeds (24.4 FPS). This is an advantage for deployment on resource-constrained platforms, making it more practical than competing methods that rely on large, slow models.
3.	The paper provides a comprehensive set of ablation studies that systematically validate the contribution of each key component. This strengthens the claims and provides clear insight into what makes the framework effective.

**Weaknesses:**

1.	The framework's performance appears to be highly dependent on a number of key hyperparameters, particularly the short (s) and long (l) temporal window sizes, the TAD margin (β), and the causal filter window size (m). The optimal values for these parameters vary significantly across different datasets (as shown in Fig. 6 and Table 12), suggesting that extensive and careful tuning is required for new tasks. The lack of a principled method for selecting these parameters could limit the model's practical applicability.
2.	The error detection mechanism relies on a majority vote across four specific pairs of predictions. While the ablation study justifies the voting threshold, the choice of the four pairs themselves feels somewhat arbitrary. A clearer rationale for why these specific pairs are optimal for capturing inconsistencies would strengthen the paper's design principles.

**Questions:**

1.	Could you elaborate on the methodology for selecting the optimal values for the key hyperparameters (s, l, β, m)? Given their sensitivity, is there a more principled or automated approach to adapt the framework to a new dataset beyond an exhaustive grid search?
2.	Regarding the majority voting scheme, could you provide the rationale for selecting the specific four pairs? Have you analyzed the contribution of other potential pairs to the final detection performance?
3.	The model is trained only on "normal" (error-free) videos. How would the system handle novel but correct variations of a procedure that were not present in the training data? Is there a risk that such unseen but valid actions would be flagged as errors due to the induced prediction inconsistency?

**Details Of Ethics Concerns:**

None.

---

> ### Author Response · Authors · 2025-11-19
> **Reviewer z56d Rebuttal**
>
> We sincerely thank the reviewer for the comments.
>
> - **Hyperparameter Selection**: We follow two guiding principles when selecting hyperparameters for the two dataset types.
>     - The first type includes datasets with many short, fine-grained actions. For these datasets such as EgoPER we primarily tune $s$ and $l$, because $\beta$ has minimal impact (as shown in Figures 5 and 10), and we set $m$ to the number of frames corresponding to a 3-second window for temporal smoothing. For $s$ and $l$, their window sizes should typically cover 1–2 actions and 3–4 actions, respectively, as suggested by Figures 13, 14, and 15. This ensures that the past-context information remains distinguishable without exceeding the GRU’s modeling capacity.
>     - The second type includes datasets where actions are long and internally composed of many fine-grained actions, such as Assembly-101-O and Epic-Tent-O. In this setting, the window sizes for $s$ and $l$ should generally cover approximately one and two actions, respectively (see Figure 12), since each action already contains multiple fine-grained actions. For this dataset type, we do not have a principled strategy for selecting $\beta$ as the number of fine-grained actions can vary substantially across videos, and we determine $m$ using the same approach as described above. Due to the large variation and complexity across datasets, an automated procedure for finding optimal hyperparameters remains an open problem, which we identify as a promising direction for future work (lines 475–477).
>
> - **Rationale for Selecting Pairs**: For ($a^s_t$, $\hat{b}^s_t$) and ($a^l_t$, $\hat{b}^l_t$), we examine inconsistencies between robust and sensitive predictions that use the same amount of past information. These inconsistencies highlight spatiotemporal differences that may reveal execution errors (e.g., placing a tortilla on a cutting board versus a table) as discussed in lines 266–269. In contrast, for ($a^s_t$, $a^l_t$) and ($\hat{b}^s_t$, $\hat{b}^l_t$), we analyze inconsistencies between predictions using short and long past-context windows. These differences emphasize changes in past contextual information, helping detect procedural errors (e.g., missing spread peanut butter before spread jelly) as mentioned in lines 293–296. Based on these considerations, we select ($a^s_t$, $\hat{b}^s_t$), ($a^l_t$, $\hat{b}^l_t$), ($a^s_t$, $a^l_t$), and ($\hat{b}^s_t$, $\hat{b}^l_t$) as the pairs that best align with our rationale for detecting both execution and procedural errors.}
>
> - **Novel but Correct Variations of a Procedure**: Our work follows the same problem setting as prior error detection studies (a one-class classification scenario in which only normal videos are available during training). We assume that the training videos cover all valid execution sequences of actions consistent with a task graph or recipe. In this setting, variations of an action are treated as errors, following the definition used in EgoPER. For example, the action “stir mug using spoon” may appear in the training data, while its variation “stir mug using knife”, although still capable of completing the task, is categorized as a modification error in EgoPER.

---

### Meta-Review · Area_Chair_8U1u · 2026-01-01

**Summary:**

The submission received mixed evaluations, including three negative scores and one positive score.

- Reviewer z56d notes that the proposed method is sensitive to multiple hyperparameters and finds the rationale behind the majority voting mechanism insufficiently explained.

- Reviewer c1rb highlights several presentation issues and raises concerns similar to those of Reviewer z56d.

- Reviewer Psmx is primarily concerned about the absence of a comparison with a recent work, MistSense, and suggests discussing the feasibility of developing LLM-based online mistake detection methods.

- Reviewer 79aK raises similar concerns that the method is not well justified (rationale and real-time property) and requires careful hyperparameter tuning. This reviewer also suggests additional related work.

**Reviewer Concerns:**

After reviewing the rebuttal, the reviews, and the manuscript, the Area Chair (AC) concludes the following:

- Hyperparameter tuning: The authors acknowledge that the method requires dataset-specific hyperparameter tuning. However, this response may not sufficiently address the reviewers’ concerns.

- Method rationale: Multiple reviewers believe that the proposed method is overly complex, with several components and design choices that are not clearly motivated or explained. Although the authors responded to these points individually, the AC believes that these concerns are unlikely to be fully resolved without substantial revisions to the manuscript that more clearly articulate the core contribution and its underlying rationale.

- Missing references and comparisons: Adequately addressed.

- Presentation issues: Can be addressed in the camera-ready version.

- Real-time property: Adequately addressed.

**Reviewer Scores:**

All reviewer scores will remain unchanged.

---

### Decision · Program_Chairs · 2026-01-26

Reject